# Combined Second Harmonic Generation and Fluorescence Analyses of the Structures and Dynamics of Molecules on Lipids Using Dual-Probes: A Review

**DOI:** 10.3390/molecules27123778

**Published:** 2022-06-11

**Authors:** Yi Hou, Jianhui Li, Bifei Li, Qunhui Yuan, Wei Gan

**Affiliations:** 1Shenzhen Key Laboratory of Flexible Printed Electronics Technology, School of Science, Harbin Institute of Technology (Shenzhen), University Town, Shenzhen 518055, China; 19b325022@stu.hit.edu.cn (Y.H.); 20b358012@stu.hit.edu.cn (J.L.); 19b958029@stu.hit.edu.cn (B.L.); 2School of Chemistry and Chemical Engineering, Harbin Institute of Technology, Harbin 150001, China; 3Shenzhen Key Laboratory of Flexible Printed Electronics Technology, School of Materials Science and Engineering, Harbin Institute of Technology (Shenzhen), University Town, Shenzhen 518055, China; yuanqunhui@hit.edu.cn

**Keywords:** second harmonic generation, two-photon fluorescence, lipid membrane, dual-probes

## Abstract

Revealing the structures and dynamic behaviors of molecules on lipids is crucial for understanding the mechanism behind the biophysical processes, such as the preparation and application of drug delivery vesicles. Second harmonic generation (SHG) has been developed as a powerful tool to investigate the molecules on various lipid membranes, benefiting from its natural property of interface selectivity, which comes from the principle of even order nonlinear optics. Fluorescence emission, which is in principle not interface selective but varies with the chemical environment where the chromophores locate, can reveal the dynamics of molecules on lipids. In this contribution, we review some examples, which are mainly from our recent works focusing on the application of combined spectroscopic methods, i.e., SHG and two-photon fluorescence (TPF), in studying the dynamic behaviors of several dyes or drugs on lipids and surfactants. This review demonstrates that molecules with both SHG and TPF efficiencies may be used as intrinsic dual-probes in plotting a clear physical picture of their own behaviors, as well as the dynamics of other molecules, on lipid membranes.

## 1. Introduction

A clear understanding on the interactions between molecules and lipid membranes is highly desired because it is the basis for biology and medical researches, including the formation of cells, the design and application of drugs, etc. [1,2]. To fulfill this purpose, multiple experimental techniques have been applied, such as circular dichroism [3,4,5], fluorescence [6,7,8,9,10], surface-enhanced resonance Raman scattering [11,12,13,14], nuclear magnetic resonance [15,16,17,18], X-ray scattering [19,20,21], SHG [22,23,24,25,26,27,28,29,30], and sum frequency generation vibrational spectroscopy (SFG-VS) [31,32,33,34,35,36,37,38,39,40,41,42]. In these studies, second order nonlinear spectroscopy, including SHG and SFG-VS, was a recently developed technique with unique interface sensitivity and exclusive interface selectivity. Under the electric dipole approximation, coherent SHG and SFG-VS may be detected only at interfaces with the broken of inversion symmetry [43,44,45,46]. For molecules in the bulk phase with inversion symmetry, second order nonlinear scatterings cannot be coherently summed and efficiently detected. For interfacial molecules with preferential orientation, SHG and SFG-VS methods may be efficiently applied to probe them. Based on this principle, SHG was generally applied in probing the asymmetry of interfaces as illustrated in Figure 1.

During the probing of interfaces in various materials, SHG was sometimes applied with the aids of other spectroscopic techniques. For example, SHG was combined with bright-field transmission microscopy (TM) to study the mechanism in Hans Christian Gram’s famous staining protocol that is generally used to differentiate bacteria [48]. The bulk molecular concentrations obtained from TM measurements can be used to estimate the molecular density on cell membranes. Gan, W. et al. combined SHG and two-photon fluorescence (TPF) to reveal that thiol adsorption on metallic nanoparticles led to annealing of surface defects that quenches photoexcitations [46]. The time lag between changes in the SHG and TPF signals was satisfactorily explained by a quantitative model. Xue, S. et al. also combined SHG and TPF to study the surface redox reaction of Ag nanoparticles in colloids [49]. Khoury, R. A. et al. combined SHG and extinction spectroscopy to monitor the seed-mediated growth of gold nanoparticles [50]. The combined spectroscopic methods provided information pointing to a two-step growth process of the nanoparticles. Dalchand, N. et al. have also combined SHG with another second order nonlinear spectroscopy, SFG-VS to investigate the mechanism of polycation−membrane interactions [51].

It may be noticed that fluorescence, including one-photon excited fluorescence and TPF, is also a common method used to study molecular interactions. For example, the partition coefficients, the dynamic behaviors, and the locations of the concerned molecules on lipid membranes were studied by measuring the changes in their fluorescence scattering [52,53,54,55]. TPF was also intensively applied as a membrane imaging method due to its predominant advantages, including longer-wavelength excitation and less photobleaching [56]. Upon excitation by laser at frequency ω, besides the SHG signal at frequency 2ω, “blue-shifted” photons of the TPF signal can also be obtained [57,58]. Therefore, it is natural and convenient to collect SHG and TPF signals generated by samples with both SHG and TPF efficiencies.

In this contribution, several reports that are mainly from our lab on the application of combined SHG and TPF methods to investigate lipids surface were reviewed. Firstly, a study on the dynamic behaviors of an anticancer drug, doxorubicin (DOX, with the structure shown in Figure 1) on the surface of vesicles prepared from a model lipid, 1,2-dioleoyl-sn-glycero-3-phospho-rac-(1-glycerol) sodium salt (DOPG) was reviewed. The general procedure in analyzing the structures and dynamics of the dual-probe molecules on lipids was illustrated in the details. It was also shown that the TPF efficiency of DOX may be modulated in a wide range by changing its molecular density on the membrane surface. Then, we showed that the combined SHG and TPF methods were used to investigate the interactions between DOPG lipid membrane and anthracyclines with different structures (Figure 1), including mitoxantrone (MIT), daunorubicin (DNR), and idarubicin (IDA). Finally, the application of SHG and TPF in studying the adsorption and transportation of dye molecules, including styryl membrane probes (FM 2–10 and FM 4–64) and 4-(4-diethylaminostyry)-1-methyl-pyridinium iodide (D289, shown in Figure 1), on the surface of vesicles, cells, and bacteria, as well as an investigation on the formation mechanism of vesicles with DOX as dual-probes, were reviewed.

## 2. Understanding and Modulating the SHG and TPF Emission from DOX on DOPG Lipids

DOX is a commonly used anti-cancer drug that is loaded into drug deliver vesicles to decrease its cardiotoxicity during medical treatment [59,60]. Understanding the structure and dynamics of DOX on the surface of both model/artificial and genuine cells is crucial for its application. At the same time, its anthracycline moiety with a large π-conjugation structure leads to an enhanced TPF efficiency upon excitation with laser at around 800 nm wavelength, as shown in Figure 2. Its structure with no inversion symmetry also leads to a detectable SHG emission upon laser excitation. For this reason, DOX was used as a good dual-probe molecule in the combined SHG and TPF investigations [57].

Figure 2 is the measured emission spectra from DOX solution, DOPG vesicle solution, and their mixed solution with the setup shown in Figure 3a [57]. With the addition of DOX in DOPG vesicle solution, TPF signal from the DOX molecules dropped significantly, while SHG signal at 400 nm position was generated because of the increased asymmetry of the DOPG lipid membrane as illustrated in Figure 1. Therefore, time-dependent SHG and TPF signals were recorded and used to analyze the molecular dynamics of DOX on lipid membranes [57].

The setups used to simultaneously monitor SHG and TPF signals are shown in Figure 3b [58,61]. By inserting a 490 nm long pass dichroic mirror in the light pass, the SHG and TPF signals at different wavelength positions were separated and detected by two photomultiplier tubes. In the reports reviewed in this manuscript [47,57,58,61,62] the SHG signal was recorded at a relatively large spatial angle range (approximately ± 40°) centered at the forward direction of the laser because of the following reason. The vesicles involved in these reports were prepared using hydration method with sonication (referred as mixed unilamellar vesicles, MUVs). Although the number mean diameters of such vesicles were approximately 100 nm, there were a small amount of relatively large vesicles in the sample which contributed the majority of the SHG scattering [63]. For this reason, the SHG scattering from MUVs was observed at a spatial angle of 5–15° away from the forward direction during the experiments. The spatial angle range of ± 40° covered most of the scattered SHG signals from such vesicle samples. It was also reported that for vesicles with number mean diameters also approximately 100 nm, which were prepared using hydration method with extrusion (small unilamellar vesicles, SUVs), the SHG scattering was at 30~35° away from the forward direction because of the absence of contribution from relatively large vesicles. Therefore, for the extrusion-induced vesicles, it is better to record their scattered SHG signal at a spatial angle around 30~35°. This report [63] also compared the SHG scattering from vesicles of three different sizes, including GUVs (giant unilamellar vesicles), MUVs, and SUVs, and observed a significant size effect on the scattered SHG intensities from these vesicles, i.e., the SHG scattering increased drastically with the increase of the vesicle diameter from ~100 nm to beyond a micron, similar to the SHG scattering from polystyrene particles discussed in a previous report [64] It was revealed that the surface curvature of vesicles influenced the permeability of the lipid membranes. Vesicles with smaller diameter tended to have more rigid lipid structures with lower permeability for a small amphiphilic molecule (D289) [63]. At the same time, our recent work also confirmed the general existence of molecular behaviors, including the adsorption, orientation flipping, embedding, and cross-membrane transportation of small amphiphilic molecules on the surface of vesicles with different sizes [65].

Figure 4 and Figure 5 show the time-dependent SHG and TPF scattering recorded after the mixing of DOX and DOPG vesicles [57]. In these figures, TPF intensity was used to reflect the fluorescence efficiency of DOX molecules while the electric field of the SHG intensity was used to reflect the orientational order of the DOX molecules. It is known that SHG field is in proportion to the surface density of net oriented molecules, i.e., the asymmetry of interfaces. The increasing of the SHG field upon the addition of DOX was naturally interpreted by the scenario that the adsorption of DOX molecules broke the interfacial symmetry of the lipid membrane. While the different changes in the TPF signal were more complicated. A physical picture as shown in Figure 6 was introduced to illustrate the structural evolution of DOX on lipid membrane and the corresponding SHG and TPF evolution [57].

Figure 6a,c were used to illustrate the adsorption of DOX on DOPG lipids at relatively low and high concentrations, respectively [57]. It demonstrated that at a relatively low surface DOX density, the adsorbed DOX molecules were separated from each other and emitted TPF efficiency equal to or slightly higher than that from free DOX molecules in solutions. At the same time, their weak orientational order prevented them to emit a detectable SHG signal. On the contrary, at a relatively high surface DOX density, the adsorbed DOX molecules aggregated and emitted a much weaker TPF signal. At the same time, their higher orientational order enabled them to emit detectable SHG signal. These changes in the SHG and TPF efficiencies were satisfactorily reflected by the initial part (in 2–3 s after DOX addition) of the recorded time-dependent curves shown in Figure 4 and Figure 5. After the adsorption, the interactions between the DOX molecules and DOPG lipids caused the structural and orientational evolutions of the DOX molecules as shown in Figure 6b,d. The embedding of DOX molecules in the lipids led to their increased orientational order. Therefore, an increase in the SHG curves was observed. Because of the above analyses, Figure 4 showed exponential and double exponential trends for the SHG field curves recorded at low and high concentration range, respectively. For lipids surface with relatively high DOX density, the embedding of DOX molecules in the lipid was associated with the separating of DOX aggregates, thus an increase in the TPF signal was observed. In experiments with the highest DOX concentration, this TPF increase was absent. This was attributed to the aggregating of DOX molecules even in the lipid bilayers.

The surface density dependent TPF signal of DOX molecules was confirmed in a subsequent study on modulating the TPF efficiency of DOX molecules by changing its surface density on DOPG vesicles [61]. As shown in Figure 7a, the addition of DOPG vesicles to DOX solution at point A significantly reduced the TPF emission because of the aggregation-caused quenching (ACQ) effect [57,66,67]. Then, with the separating of the DOX molecules aggregated on vesicle surface, induced by the addition of more vesicles at points marked as B to J, the TPF emission per molecule gradually increased. Assuming the DOX molecules in the sample are all distributed on the vesicle surface, the TPF efficiency of DOX on the lipid surface with the apparent mean distance between nearby DOX molecules was calculated and plotted (Figure 7b). This work clearly demonstrates that by altering the chemical environment of DOX molecules, their TPF efficiency may vary in the range of ~4% to ~300% compared with the TPF efficiency in their aqueous solution [57,61].

In the aforementioned TPF analyses, the quenching of the TPF efficiency from the known ACQ effect [68,69] was the key for estimating the number density of DOX molecules on DOPG lipids. For example, a quenching of the TPF scattering to 4% of its original value (the TPF signal in aqueous solution with no DOPG vesicles) leads to two conclusions: one is that the partition coefficient of DOX molecules on the DOPG lipids is higher than 96%, another is that the TPF efficiency of the aggregated DOX molecules is less than 4% of their TPF efficiency in aqueous solution. Both the statements need to be true to achieve a 96% decrease in the TPF scattering. It was also revealed that for DOX, one-photon fluorescence had a similar change with TPF [57]. That is to say, the combined analyses may also be performed with SHG and one-photon fluorescence measurements. However, the experimental setup shown in Figure 3 provides an easy way to detect the time-dependent SHG and TPF scattering at the same time, which enables a better choice of SHG and TPF analyses.

Besides the application in estimating the interfacial density of DOX molecules on DOPG vesicles, the TPF signal was also used to reflect the dynamics of the behaviors of DOX molecules on lipids. The time constants in fitting the SHG and TPF evolution curves in Figure 4 and Figure 5 were shown in Figure 8 [57]. The fast change in the SHG field curves (*τ*_1_) was only observed for experiments with relatively high DOX concentrations, while the relatively slow SHG change (*τ*_2_), which was attributed to the embedding of DOX in DOPG lipids, was observed in all the experiments, in line with the physical pictures illustrated in Figure 6. The separating of the DOX aggregates (*τ*_3_) occurred at a time scale between *τ*_1_ and *τ*_2_, which is also reasonable.

## 3. Influence of the Molecular Structure on Dynamic Behavior of Anthracyclines on Lipids

As another example of applying the combined SHG and TPF analyses, an investigation on the structure and dynamics of anthracyclines with different side groups, including MIT, DNR, and IDA, revealed the influence of molecular structure on their dynamics on DOPG lipid bilayer [58]. As shown in Figure 9b,c, all three anthracyclines had detectable SHG emissions upon their adsorption on DOPG lipids. DNR and IDA were also applied as dual-probes similar to DOX because their TPF efficiencies were observed to be sensitive to the chemical environment. However, MIT was only used as an SHG-probe because its TPF emission was too weak to be analyzed.

The time-dependent evolution of the SHG and TPF scattering from the experiments shown in Figure 9 and the experiments with 20 μM DOX shown in Figure 4 and Figure 5 were plotted in Figure 10. These data reveal the dynamic behavior of the four drugs on the surface of DOPG vesicles [58]. Firstly, the four SHG curves leveled off after tens to hundreds of seconds, indicating that all four anthracycline types of anti-cancer drugs distributed on the surface of DOPG vesicles and did not translocate across the lipid bilayer at the concentration used in the experiments. This observation was somewhat unexpected because it is known that these drugs may enter cells during medical treatments. It was, however, revealed that these drugs penetrated across the bilayers of DOPG lipids at relatively high concentrations in aqueous solution or in other solutions including PBS and salt solutions [29,57,58,70,71]. Secondly, three anthracyclines with detectable TPF emission (DOX, DNR, and IDA) presented similar trends in their SHG and TPF curves. Analyses of the SHG and TPF curves revealed their similar dynamic behaviors, including the adsorption and aggregation, then de-aggregation and embedding in the lipids. The different trend in the SHG curves recorded from MIT experiments was also interpreted by the different MIT adsorption structures as shown in Figure 11. During embedding of the four anthracyclines in the lipid bilayer, MIT flipped its orientation thus a decreasing in the SHG field curves was observed. The other three anthracyclines embedded in the lipids with their orientations kept the same direction, thus only an increasing in the SHG filed curves induced by better orientational order was observed.

## 4. Investigating the Vesicle Formation Process and Dynamics of Dye Molecules on Lipids and Cells

The combined SHG and TPF analyses were also performed with the aid of Rayleigh scattering in studying the self-assembly of DOX and surfactant, sodium bis (2-ethylhexyl) sulfosuccinate (AOT) and the formation of binary complex vesicles in solution [47]. In this report, the time evolution of the SHG, TPF, and Rayleigh scattering signals plotted in Figure 12 were used to propose a formation mechanism of the DOX/AOT vesicles illustrated in Figure 13.

As aforementioned, SHG signal reflects the asymmetry of the interfaces, including the interfaces of the possible structures in the mixed solutions including the micelles, aggregated micelles and vesicles. At the same time, the size of these structures is crucial for their SHG efficiency [47,63,64]. For this reason, the increase in the SHG signal was used as an indicator of the formation of aggregated micelles and the transformation from aggregated micelles to vesicles. On the other hand, the decrease in the SHG signal was an indicator of the cross-membrane transport of DOX molecules from outside the vesicle surface to the inside leaflet of the lipid membrane.

The size evolution of the structures in the mixed solution was confirmed by the Rayleigh scattering data. Because Rayleigh scattering is determined by the equation of $i_{s\left(\theta \right)}=I_{0}\frac{9\pi^{2}V^{2}N}{2r^{2}\lambda^{4}}{\left(\frac{n^{2}-n_{0}{}^{2}}{n^{2}+2n_{0}{}^{2}}\right)}^{2}\left(1+\cos^{2}\theta \right)$ (*I*_0_ is the incident intensity; *V* and *N* are the volume and number of the structures in the solution, respectively; *r* is the distance from the sample to the detector; *λ* is the wavelength of the light; *n* and *n*_0_ are the refractive index of the structures and the solvent in the colloids, respectively; θ is the scattering direction), the increased size of the structures in the solutions led to an increase of $i_{s\left(\theta \right)}$. Furthermore, during the structural evolution as shown in Figure 13, the aggregating of the DOX molecules and their separating in the surface of formed vesicles were also confirmed by the change of the TPF efficiency of DOX molecules.

This formation mechanism of the DOX/AOT binary complex vesicles was also confirmed by the dynamic light scattering measurements and transmission electron microscope images. This work [47] presented a new formation mechanism for the self-assembled complex vesicles with the absence of floating lipid bilayers in the solution as precursors. The rationality behind this deduction is that floating lipid bilayers in the solution tend to have symmetric structures and form symmetric vesicles directly. The observation of the asymmetric vesicles as shown in Figure 13 based on the notable decreasing in the SHG curves ruled out the existence of such precursors.

The formation of vesicles by the generally applied hydration method from stacked DOPG bilayers, as proposed in Figure 14, was also investigated by the combined SHG, TPF and Rayleigh scattering approaches. This formation process in Figure 14 was generally accepted in previous reports [72,73,74]. The combined multiple spectral methods were demonstrated to be capable of revealing the detailed structural evolution of the lipids during this process [47].

The stripping of the stacked bilayers (shown in Figure 14A) and the formation of RTLS (relatively thick lipid structures, Figure 14B), the swelling of curved lipid bilayers on RTLS (Figure 14C), and the formation of vesicles (Figure 14D) were precisely followed by the Rayleigh scattering curves as shown in Figure 15. The generating of RTLS in the solution caused a significant increase in the Rayleigh scattering curves and the transformation from RTLS to vesicles caused a decrease. This structural evolution of the lipids was confirmed by the SHG and TPF with DOX as a dual-probe. The adsorbed DOX molecules caused asymmetric interfaces in structures B, C, and D, which was reflected by the SHG curves. At the same time, the aggregating and then separating of DOX molecules were also reflected by the decreasing and the followed increasing of the TPF curve.

In this investigation [47], the number of DOX molecules aggregated on the interfaces was estimated by the quenching of the TPF signals. Furthermore, the distribution of the interfacial DOX molecules on the interfaces was also estimated by the change in the SHG intensities. For example, the asymmetric vesicles shown in Figure 13 had proximately 75% of the adsorbed DOX molecules outside the vesicles and 25% inside the vesicles. For the vesicles as shown in Figure 14, no DOX molecules were distributed inside the hollow structures during the swelling (C structure), thus there were no DOX molecules inside the DOPG vesicles. The dynamics in the structural evolutions in Figure 13 and Figure 14 were also satisfactorily verified by the matching of the time-evolution curves in Figure 12 and Figure 15, respectively. These details were presented in the report [47], thus omitted here.

Being the opposite to the ACQ effect observed for the aforementioned anthracyclines, some molecules were revealed to emit enhanced TPF signal upon their aggregating and embedding on/in the lipid bilayers [62,75,76]. This property is also helpful in analyzing its interfacial behaviors on lipids. Miller et al. studied the interaction between styryl molecules (FM 2–10 and FM 4–64) and two different Gram-positive bacteria by collecting the SHG and TPF signals simultaneously [76]. They observed that the TPF and SHG signals both increased in the adsorption of FM 4–64 on bacteria, as shown in Figure 16. The decrease in the SHG signal for experiments with *Staphylococcus aureus* (*S. aureus*) (Figure 16a) was attributed to the transportation of FM 4–64 molecules from the outer leaflet to the inner leaflet of the cell membrane. The absence of such a decrease in the SHG signal of *Enterococcus faecalis* (*E. faecalis*) experiments (Figure 16c) was used to reveal a more rigid cell membrane for *E. faecalis*. The TPF curves shown in Figure 16b,d were also used to confirm this interfacial dynamics with the decrease in TPF attributed to the photobleaching effects [76]. The adsorption isotherms of the FM dyes on the bacteria and their dissociation constants were also estimated by fitting the TPF and SHG isotherms with the Langmuir adsorption model [76].

The dynamics of D289 on the surface of human chronic myelogenous leukemia (K562) cells and the sub-cellular structures was investigated by the combined SHG and TPF spectroscopy and SHG imaging methods [62]. As shown in the left part of Figure 17, SHG and TPF curves all showed a notable increase with the addition of D289 to the K562 cells. After this rapid increasing, the TPF curves kept increasing with time, while the SHG curve first decreased then increased. These changes in the SHG and TPF signals were used to estimate the dynamics of the transportation of D289 molecules in the K562 cells, and the distribution of D289 on the sub-cellular structures, such as mitochondria. Such interactions between D289 and a K562 cell was also confirmed by the successively recorded SHG images shown in the right part of Figure 17. The relative fast transportation of D289 molecules into mitochondria was also revealed by the SHG and TPF measurements [62].

These reports demonstrated that the enhanced TPF signals may be applied to reveal the dynamic behavior of molecules with chromophores on interfaces [62,76]. However, it was not possible to use the enhanced TPF signal to estimate the exact number of molecules located on the interfaces because the ratio of the TPF enhancement was unknown.

## 5. Summary and Outlook

The reviewed works [47,57,58,61,62] demonstrate that with the introducing of dual-probes, the combined SHG and TPF techniques provide valuable information on the structures and dynamics of molecules on lipid membranes. Besides the interfacial asymmetry probed by the SHG technique, the quenching of the TPF scattering from the ACQ molecules may provide information on their interfacial densities. The time-evolution of the SHG and TPF signals can also be compared to obtain clear dynamic behaviors of molecules on interfaces. During this approach, the quenching of the fluorescence is crucial, but it may prevent the application of dual-probes with relatively low TPF efficiency. Therefore, the enhancement of the fluorescence from probing molecules in their changed chemical environment, such as the well-known AIE effect [77,78], warrants further investigation. At the same time, improving the sensitivity of the signal detecting system is always helpful for extending the choice of dual-probes and the application of the combined analyses methods.

It has been demonstrated that the combined spectroscopic method is also applicable in studying molecular dynamics on metallic nanoparticles [46,49]. Possibly, more applications of the combined analyses can be revealed in the near future. Microscopic investigations may provide more information on the localized effect on the structures and dynamics of molecules on lipids. The combined SHG and TPF imaging setup has been developed and applied in many biology studies [79,80,81,82,83]. The approaches demonstrated in this contribution may be directly applied in analyzing the SHG and TPF images recorded during the adsorption and cross-membrane transportation of dual-probes on/in vesicles and cells to resolve puzzles in medical and biophysical studies.

## Data Availability

Not applicable.

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
