# Peer review of "Combined Second Harmonic Generation and Fluorescence Analyses of the Structures and Dynamics of Molecules on Lipids Using Dual-Probes: A Review"

_molecules, 2022, doi:10.3390/molecules27123778_

Round 1
Reviewer 1 Report
"The manuscript submitted by Yi Hou , Jianhui Li, Bifei Li, Qunhui Yuan and Wei Gan, describe the application of combined SHG and TPF on the study of the dynamics of molecules on lipids, using dual probes. This article gives a reasonable review on this topic and the authors own perspective on this area. The review is well written and well organized. I believe it is of interest to the audience of Molecules. In my opinion it may be published after minor corrections
Please find my comments below:
Line 17. “Fluorescence emission, which is in principle not interface selective but varies with the chemical environment where the chromophores locate, was also revealed to be capable of revealing the dynamics of molecules on lipids.” Please correct the sentence
Line 62- “The information obtained from the SHG and extinction spectroscopies was complementary and they pointed to a two-step…” Please correct the sentence
Line 255. In my opinion, the subtitle 3 “ Anthracyclines with different structures behave differently on lipids” should be changed. “Influence of the molecular structure of Anthracyclines on lipids dynamics” is proposed.
Line 387. “S. Aureus” and “ E. Faecalis” should be in italic.
Reviewer 2 Report
The comments are as follows: To overcome the existing barriers to molecular analysis in biology, and medicine, we need to develop methods with the performance capability of rapid and reliable analysis at the molecular level. Usually, the most common analytical techniques used in analyses of lipids and lipidomes suffer from some vantages and weaknesses and these techniques must be used in combination with other techniques. Among these analytical methods, matrix-assisted laser desorption/ionization mass spectrometry meets these requirements; however, some limitations complicate its application to analyze the small molecules. In this contribution, the authors provide a review of their recent works focusing on using of the combined analytical methods, i.e., SHG and two-photon fluorescence (TPF). They believe that the combination of both SHG and TPF analytical methods may be considered as intrinsic dual-probes in plotting a clear physical picture of molecular behaviors, as well as the dynamics of other molecules, on lipid membranes. It seems that the finding of this study can provide valuable information for further improvement of the intrinsic dual-probes for molecular analysis.
The manuscript can be accepted for publication without any substantial modification.
Sincerely Morteza
